# Breast Milk Content of Vitamin A and E from Early- to Mid-Lactation Is Affected by Inadequate Dietary Intake in Brazilian Adult Women

**DOI:** 10.3390/nu11092025

**Published:** 2019-08-29

**Authors:** Michele R. Machado, Fernanda Kamp, Juliana C. Nunes, Tatiana El-Bacha, Alexandre G. Torres

**Affiliations:** 1Laboratory of Food Science and Nutritional Biochemistry, Institute of Chemistry, Federal University of Rio de Janeiro, Rio de Janeiro 21941-909, Brazil; 2School of Nutrition, Faculdade Arthur Sa Earp Neto (FMP/FASE), Petropolis, Rio de Janeiro 25680-120, Brazil; 3Biochemistry Core, Federal Institute of Education, Science and Technology of Rio de Janeiro, Rio de Janeiro 20270-021, Brazil; 4School of Nutrition, Federal University of the State of Rio de Janeiro, Rio de Janeiro 22290-250, Brazil; 5Institute of Nutrition, Federal University of Rio de Janeiro, Rio de Janeiro 21941-902, Brazil; 6Department of Physiology, Development and Neuroscience, University of Cambridge, Cambridge CB2 3EL, UK

**Keywords:** retinol, α-tocopherol, inadequate intake, nutritional status, breast milk, undernourishment, dietary assessment, multiple source method

## Abstract

Our aims were to investigate vitamin A and E status during lactation and the determinants of breast milk content for the appropriate nutrition of the infant in a study with nursing Brazilian women. We hypothesized that both inadequate intake and the lipoprotein distribution of vitamin A and E during lactation could have an impact on their breast milk levels from early- to mid-lactation. Nineteen adult lactating women participated in this longitudinal observational study, in which dietary records, blood and mature breast milk samples were collected for the analysis of vitamin A and E, and carotenoids in early- (2nd to 4th week) and mid-lactation (12th to 14th week). Nutrient intake was balanced by the Multiple Source Method (MSM), and the intake of vitamin A and E was inadequate in 74 and 100% of the women, respectively. However, these results were not reflected in low serum concentrations of retinol and only 37% of the volunteers were vitamin E deficient according to the blood biomarker. As lactation progressed, vitamin A and E status worsened, and this was clearly observed by the decrease in their content in breast milk. The reduced content of vitamin A and E in the breast milk was not related to their distribution in lipoproteins. Taken together, the contents of vitamin A and E in breast milk seemed to be more sensitive markers of maternal nutrition status than respective blood concentrations, and dietary assessment by the MSM in early lactation was sensitive to indicate later risks of deficiency and should support maternal dietary guidance to improve the infant’s nutrition.

## 1. Introduction

Vitamin A and E are essential to newborns, and their transfer to breast milk is key for the nutrition of infants. Vitamin A is crucial for newborn development, epithelial function and protection against infections. Vitamin A deficiency affects millions of preschool-age children, especially in developing countries, and it increases the susceptibility to infection and infant mortality rate, especially before the age of 2 [1]. Vitamin A nutritional requirements are met by the intake of retinoids, such as retinyl esters, and provitamin A carotenoids, especially β-carotene and less prominently by α-carotene and β-cryptoxanthin. In addition to serving as retinoid precursors, these carotenoids are transferred to breast milk and, together with lycopene and lutein, might contribute to breastfed infant health, as they present potential bioactivity. The primary function of vitamin E is related to its activity as a potent free radical scavenger and, although overt signs of deficiency are rare, pre-term infants might be more susceptible to associated illnesses [2].

Essential nutrients, such as vitamins, are provided to the breastfed infant via transfer from the maternal circulation into the milk by the mammary gland. The contents of retinol and α-tocopherol in the breast milk are mainly determined by maternal diet and nutritional status [3,4]. However, to what extent changes in the maternal nutritional status throughout lactation might affect the transfer of vitamin A and E to the breast milk is not entirely known. Interactions between vitamin A and E seem to influence their contents in breast milk as it has been shown that postpartum retinyl-ester supplementation decreased vitamin E content in the colostrum [5]. On the other hand, the mechanisms underlying this biochemical interaction are not known and are possibly related to serum transport in lipoproteins and uptake by the mammary gland [3]. Vitamin A and E distribution in major lipoprotein fractions might fluctuate throughout lactation, which would impact their transfer to breast milk.

Our aim was to investigate vitamin A and E status, and carotenoids during lactation, and the determinants of breast milk levels for the appropriate nutrition of breastfed infants. We hypothesized that both inadequate intake and the lipoprotein distribution of vitamin A and E during lactation could have an impact on their breast milk levels from early- to mid-lactation. By taking advantage of a longitudinal study with adult nursing women in Rio de Janeiro, Brazil, we used multiple regression models to explain the impact of maternal diet and serum concentration of vitamin A and E on their breast milk content during early- and mid-lactation.

## 2. Materials and Methods

### 2.1. Experimental Design and Recruitment of Volunteers

This is a longitudinal observational study, approved by the Research Ethics Committee of the Pedro Ernesto University Hospital, Rio de Janeiro State University (CEP/HUPE-UERJ:3043/2011–CAAE:0186.0.228.228-11). Data and sample collection were conducted after signing written informed consent. The volunteers were recruited at Policlínica Américo Piquet Carneiro (UERJ) and at health facilities in the city of Rio de Janeiro. Recruitment occurred in the last month of gestation, when the first visit of the study was scheduled according to the probable delivery date. Of the 22 recruited women, three refused to participate during mid-lactation and their data were excluded. Thus, 19 pregnant women (age 20–40 years) participated in the study, exclusively or predominantly breastfeeding. They did not have any chronic or acute diseases, were not using prescribed medicines or dietary supplements, and were non-smokers and non-alcoholic who had completed single and full-term gestations. In the present study, mature breast milk was collected, because in full lactation, when mature milk is produced, tight junctions between mammary epithelial cells are fully functional, and, therefore, paracellular transport (between cells) is virtually absent. Therefore, one can expect that there are no major changes in the mammary epithelial cells transport systems during the time span assessed in this study.

### 2.2. Data Collection and Qualification of Lactation Practices

Demographics and medical history were obtained in early lactation (2nd to 4th week postpartum; mean 24.7 days). In mid-lactation (12th to 14th week postpartum; mean 94 days), volunteers answered a questionnaire concerning lactation practices. All women reported adopting practices consistent with exclusive or predominant breastfeeding. Total body mass (kg) and height (meters) were obtained in an anthropometric scale (Filizola, Brazil) to calculate body mass index (BMI, in kg/m^2^) [6].

### 2.3. Assessment of Dietary Intake

Dietary intake throughout lactation was estimated using the Multiple Source Method (MSM). The MSM is a statistical method to estimate usual food intake and uses at least two different dietary inputs (e.g., dietary recalls and/or food frequency questionnaires) [7] and it has been shown to be adequate to estimate usual intake during pregnancy [8]. Using this method, 24-h dietary recalls (24hRs) were applied in early- and mid-lactation (one in each period) to evaluate possible changes in eating habits and to estimate recent intake of energy and nutrients. The nutrient composition of the 24hRs was analyzed with the NutriSurvey^®^ software v. 2007 (Dr. Juergen Erhardt; Germany; available at www.nutrisurvey.de/nutrisurvey2007.exe) using the food composition database from the US Department of Agriculture release 22 [9] adapted for common fat sources in the Brazilian diet [10]. A semi-quantitative food frequency questionnaire (SQ-FFQ) was applied in the first lactation period to assess food sources of the nutrients of interest in the present investigation and to confirm the feasibility of the 24hR to evaluate usual dietary intake.

The prevalence of inadequate nutrient intake was estimated using Estimated Average Requirements (EARs), as set by the Institute of Medicine [11,12,13,14] as cut-off points. The prevalence of the inadequate intake of each nutrient was estimated by the proportion of individuals with intake below the EAR value. Since the EAR cut-off point method is not suitable for assessing energy adequacy, the proportion of the group with BMIs below, within, and above the desirable range was used to classify inadequate, adequate, and excessive energy intakes [11].

### 2.4. Collection and Processing of Biological Samples

Blood and milk samples were collected in the morning after fasting overnight, were immediately processed for the separation of plasma, serum and erythrocytes, and frozen at −80 °C until analysis. Whole blood aliquots were used for the immediate determination of hematocrit and hemoglobin concentrations. Milk samples (10 mL) were manually expressed in the morning from the breast which the baby last suckled and transferred into sterile plastic bottles. An aliquot was taken for analysis of the crematocrit. For the analysis of lactose and protein, aliquots were stored at −20 °C and for retinol, carotenoids and tocopherols analyses, at −80 °C and tubes were protected with aluminum foil.

### 2.5. Determination of Hematocrit and Hemoglobin, and Triglycerides and Cholesterol in Blood

Hematocrit was determined in a microhematocrit centrifuge (Hemospin, Incibrás; São Paulo, Brazil) and hemoglobin concentration was determined by the cyanmethemoglobin method by a commercial kit (BioClin; Belo Horizonte, Brazil). Triglyceride, total cholesterol and HDL-c (total serum and HDL-c fraction after LDL-c+VLDL-c precipitation) were determined by the enzymatic colorimetric method using commercial kits (BioClin, Brazil), using sodium phosphotungstate and Mg^2+^ for precipitation of LDL and VLDL particles, as described [15]. The clear supernatant, containing the HDL fraction, was used for the analysis of HDL-c (and vitamins by HPLC, described in Section 2.7), and the concentration of LDL-c+VLDL-c was determined by the difference between the concentration of HDL-c and total cholesterol. All analyses were performed in duplicate and the automatic reading was performed in triplicate.

### 2.6. Breast Milk Macronutrient Composition

Total fat was determined by the crematocrit method, in a microhematocrit centrifuge (Hemospin, Incibrás) [16]. Lactose was determined by a colorimetric assay [17] adapted by [18], based on the reaction with picric acid. Total protein content was determined by the Lowry method [19], after eliminating interference of fat and lactose, by precipitating the proteins with 20% trichloroacetic acid and 2.5% sodium deoxycholate, and centrifugation. The supernatant with interfering substances was discarded and precipitated milk proteins were suspended in Folin–Ciocalteu’s reagent and analyzed [20]. The standard curve was prepared with casein in concentrations ranging from 20.0 to 100.0 μg/mL. All these analyses, as well as standard curves, were performed in triplicates.

### 2.7. Determination of Retinol, Carotenoids and Tocopherols by HPLC in Breast Milk, Whole Blood Serum, and Lipoprotein Fractions

Serum samples were prepared and extracted as described by [21], adapted from [22]. Retinol, tocopherols, and carotenoids were analyzed by HPLC in serum and in the HDL fraction, in duplicate. The concentration of these vitamins in the LDL+VLDL fraction was calculated by the difference between contents in whole serum and in the HDL fraction. Milk samples were prepared as described [21], adapted from [23], and care was taken to avoid direct light to avoid vitamin losses during sample extraction and analysis.

HPLC analysis was performed in a Shimadzu chromatographic system (Shimadzu, Kyoto, Japan), composed of a LC-20AT pump, a Shimadzu SPD-M20AV UV-Vis detector, and a CBM-20A system controller. A total of 20 μL of sample extracts and standards solutions were injected through a Rheodyne injection valve with a volumetric loop, onto a C18 reversed-phase column (Kromasil; 150 × 4.6 mm), and compounds were eluted isocratically with a ternary solvent composed of acetonitrile, tetrahydrofuran, and 15 mM methanolic ammonium acetate (65:25:10, v/v/v), at 0.9 mL/min. Each sample extract was injected twice, one run monitored at λ (nm) 450, for the analysis of carotenoids, and the other run monitored at 325 and 292 for the analysis of retinol and tocopherols, respectively. All analyses were performed in duplicate.

Identities of chromatographic peaks corresponding to the fat-soluble vitamins in samples chromatograms were determined based on standards retention times, and by exact co-elution with standards spiking in representative samples. Vitamin concentrations in samples were determined by external calibration, with linear calibration curves and direct extrapolation. Commercial standards of retinol, α-tocopherol and γ-tocopherol were from Sigma-Aldrich (São Paulo, Brazil), and carotenoid standards (purity > 95%) were isolated from foods naturally rich in these pigments, as previously described [24], by the open column method [25]. Representative chromatograms from HPLC standards are shown in Appendix A.

### 2.8. Statistical Analyses

Paired *t*-tests were used to investigate differences between early- and mid-lactation on the variables studied. Variable frequency distribution was assessed by standardized coefficients of skewness and kurtosis, and those with values < −2.0 or > +2.0 were characterized as having a non-normal distribution, which were presented as median and minimum and maximum levels, in contrast to normally distributed variables that were presented as the mean ± standard deviation (SD). Non-normally distributed variables were log-transformed before running Pearson’s correlation analysis, which was used to investigate associations between continuous variables. Stepwise multiple regression analyses (backward) were used to investigate the effect of independent factors on the composition of vitamins in serum and in breast milk. The criteria for the inclusion of independent variables in the multiple regression models were based on results from Pearson correlations and on data from the literature. In the final model, only significant variables that improved the model were kept (*p*-to-remove ≥ 0.05; *p*-to-persist < 0.05). The multiple regression models were further assessed by analysis of residual plots that were checked to determine if they were randomly distributed. Data analyses were performed with the Multiple Source Method (MSM) software for dietary data analysis, GraphPad Prism 7.0 (GraphPad Software, San Diego, CA, USA) and Statgraphics Centurion 18 (Statgraphics Technologies, Inc.; The Plains, VA, USA) for statistics. Values of *p* < 0.05 were considered significant.

## 3. Results

### 3.1. General Characteristics of Lactating Women

The nursing women who participated in the present study were adults (31.9 ± 5.9 years of age), primiparous and had term pregnancies (gestational age of 39.8 ± 1.1 weeks), from which 58% had vaginal delivery (Table 1). According to the BMI, 58% of women were overweight (25–30 kg/m^2^ [6]) in early lactation (2nd to 4th lactation week). In mid-lactation (12th to 14th lactation week), 42% of the lactating mothers were overweight. Most of the volunteers (63%) were anemic at the beginning of the study and after 10 weeks, there was a significant 11% increase in the mean blood hemoglobin concentration and a decrease in anemia frequency to 26% in mid-lactation (Table 1).

Serum triglyceride concentration was adequate in all volunteers and reduced significantly by 24% from early- to mid-lactation. Serum HDL concentrations, on the other hand, were below the minimum value in all volunteers and 74% of the nursing women presented LDL concentrations above the desirable range, irrespective of the lactation period [27]. Regarding total cholesterol in the serum, 47 and 31% of the volunteers presented concentrations above the desirable value, in early- and mid-lactation, respectively.

### 3.2. Dietary Intake of Vitamin A and E Were Inadequate throughout Lactation

Dietary data assessment indicated that nutrient intake did not vary significantly between the two lactation periods, considering both recent and usual dietary consumption. Therefore, these data were presented combined, as estimated by the MSM, which assesses habitual intake considering data from two or more 24hRs and one FFQ (Table 2). The frequency of volunteers with sub-adequate nutrient intake was estimated as proposed [12,13] using EAR whenever available (Table 2). The very high prevalence of inadequate intake of vitamin A and E (Table 2) may set these lactating mothers and their offspring’s health at increased risk. Although the median intake of β-carotene (Table 2) was within the range considered prudent, 47% of the volunteers presented β-carotene intake below this range. In contrast, the average intake of lycopene was lower than the level established as prudent [12], and intake was sub-adequate in 89% of the volunteers.

### 3.3. Serum Vitamin A and E Were Not Associated with Their Changes in Breast Milk Concentration throughout Lactation

Serum vitamin A was adequate in all volunteers in early- and mid-lactation (Table 3). In contrast, 68 and 84% of the volunteers were at risk of developing vitamin E deficiency in early- and mid-lactation, respectively. Although serum retinol concentration did not vary throughout lactation, retinol in breast milk showed a significant 13% decrease from early- to mid-lactation (Table 3). In early lactation, retinol concentration in the milk of all volunteers was above the limit considered adequate [1,3] for the formation of infant liver reserves. After 10 weeks of lactation, however, milk retinol concentration became inadequate in 21% of the nursing women [1]. A similar trend was observed with breast milk contents of β-carotene, lutein+zeaxanthin and α-tocopherol, which significantly decreased from early- to mid-lactation, irrespective of the serum concentrations (Table 3).

### 3.4. Breast Milk Concentration of Fat-Soluble Vitamins from Early- to Mid-Lactation Is Associated with Dietary Vitamin A, Serum β-Carotene and Tocopherols

Multiple regression analysis was used to investigate the determinants of vitamin A and E and carotenoid concentration in breast milk (Table 4). Retinol concentration in breast milk was determined by the dietary intake of vitamin A, and this association was stable from early- to mid-lactation (Table 4; models 1 and 4). In contrast, the determinants of milk β-carotene changed from early- to mid-lactation (Table 4; models 2 and 5). Regarding vitamin E, serum concentration was the sole determinant of its contents in breast milk (Table 4; models 3 and 6).

### 3.5. Vitamin A and E and Carotenoids Distributed Differently in Lipoprotein Fractions during Lactation

There were no significant changes from early- to mid-lactation in the relative distribution of vitamin A and E and carotenoids in serum lipoprotein fractions (Figure 1). Retinol and γ-tocopherol were distributed equally between HDL+serum binding proteins and LDL+VLDL fractions, in both lactation periods. β-Carotene and α-carotene, lycopene and α-tocopherol were enriched in the LDL+VLDL fraction. In contrast, approximately 70% of lutein+zeaxanthin was concentrated in the HDL+binding proteins fraction, in both lactation periods.

## 4. Discussion

Adequate breast milk contents of vitamins and minerals are crucial for infant development. In the present work, the maternal nutritional status of vitamin A and E was evaluated, and the factors associated with their content in breast milk in adult Brazilian nursing mothers. The most striking result was the fact that despite constant serum levels, the content of vitamin A and E in breast milk decreased throughout lactation, which we hypothesize as a consequence of the sustained inadequate dietary intake from early- to mid-lactation. This sub-adequate intake was insufficient to affect biochemical markers of vitamin status in serum, but negatively affected breast milk, being potentially detrimental to the lactating infant. Therefore, vitamin A and E in breast milk seem to be more sensitive markers of maternal nutrition status than their respective blood concentrations. Since the distribution of vitamin A and E in fasting serum lipoproteins did not change from early- to mid-lactation, this factor is unlikely to be related to the decrease in the transfer of fat-soluble vitamins into milk during lactation.

### 4.1. Dietary Intake, Nutritional Status and Milk Transfer of Vitamin A and E, and Carotenoids

Vitamin A intake was sub-adequate in 74% of the subjects [13] and was insufficient to maintain adequate breast milk levels throughout lactation. Maternal vitamin A requirement increases during lactation to fulfill the increased demand imposed by the mammary gland, and vitamin A in breast milk is essential to build up the infant’s liver stores, contributing to the prevention of vitamin A deficiency when lactation ends [4,29]. A high intake of β-carotene could compensate the low intake of retinol, but the former nutrient was in the range considered prudent (3000–6000 µg/day [12]) in only 53% of the volunteers, thus it is likely that the breastfed infants were at risk of vitamin A deficiency [4,30]. Despite that, all subjects showed serum retinol levels above the cutoff value for inadequacy risk (>0.7 µmol/L [1]) throughout lactation [3].

Retinol in breast milk is also seen as a reliable marker of vitamin A nutritional status on a population basis. According to the World Health Organization, when retinol concentration in milk is below 1.1 μmol/L, the infant’s body stores may be below the estimated critical amounts to supply for the increased requirements in the second half of childhood [1]. In all subjects in the present study, the breast milk concentration of retinol in early lactation was above this cutoff value. However, in mid-lactation, 21% of the nursing women were below the adequacy cutoff. Possibly, these results indicate that these nursing women presented subclinical vitamin A deficiency, which worsened throughout lactation, and that their lactating newborns were at risk of vitamin A deficiency to build up liver stores [4,29]. However, concentrations of fat-soluble vitamins are usually higher at the beginning of lactation, suffering a progressive fall as lactation progresses [3], but levels should be above the safe cut-off value throughout lactation.

The retinol concentration in breast milk in the two lactation periods evaluated was near the average values previously reported for Brazilian nursing women [5,21,30] and for Korean nursing women (2.0 to 3.0 μmol/L) [31]. As these previous works did not assess dietary intake, further discussion regarding the factors involved in the regulation of the concentration of vitamins in breast milk was not possible. Retinol content in breast milk is associated with maternal dietary fat intake, stressing the importance of dietary lipids to retinol transport and absorption [31]. It is suggested that during lactation, a large proportion of retinol from the maternal diet is transported directly into the mammary gland via blood lipoproteins, bypassing the liver [3]. Breast milk vitamin A might be more strongly affected by dietary fat intake than by serum retinol concentration, especially in groups with a low dietary intake of vitamin A and inadequate nutritional status of this vitamin [31]. Consistently, in the present study, fat intake was correlated with breast milk retinol, both in early (*r* = 0.47; *p* = 0.042) and in mid-lactation (*r* = 0.46; *p* = 0.046) and vitamin A intake was a determinant of breast milk retinol concentration in both lactation periods investigated (Table 4).

The intake of vitamin E was also sub-adequate in all volunteers [12]. Vitamin E intake might have been underestimated in the present study, because its principal dietary source was vegetable oil, and it is inherently difficult to assess cooking oil intake [32]. In Brazil, soybean oil is by far the most consumed oil and the main source of vitamin E in the subjects’ diets. In contrast, it should also be considered that the possibility that vitamin E EAR (16 mg/day [12]) is overestimated, which could contribute to the high frequency of inadequacy in the present study [32]. In this sense, the recommended daily intake of vitamin E for Brazilian nursing women [32] is nearly half that recommended in North America by the Institute of Medicine [4,12]. The vitamin E status biomarker (serum α-tocopherol/cholesterol ratio; µmol/mmol) indicates that the prevalence of vitamin E deficiency (<2.25 µmol/mmol) [2,4] increased from 37 to 63% of the volunteers from early- to mid-lactation.

β-Carotene was the major carotenoid present in the serum and breast milk throughout lactation, consistent with maternal usual diets, representing over 50% of total carotenoids in milk [21,33]. Serum and breast milk β-carotene were strongly associated, as expected [34,35] and, in early-lactation, breast milk β-carotene was also associated with maternal intake, highlighting the importance of both dietary intake and blood concentration to β-carotene in breast milk [34,35]. It is noteworthy that the concentration of β-carotene and lutein+zeaxanthin in breast milk was, respectively, approximately 8- and 6-fold higher than observed in a previous study with Brazilian nursing women [21]. Compared to the values of a recent study that addressed the effect of prematurity on the content of carotenoids in the colostrum, the results of the present study were sensibly lower, particularly for lutein+zeaxanthin and lycopene concentrations, which were approximately 10-fold lower [36]. These differences might be related to the stage of lactation and also to potential differences in the habitual diets of the volunteers of both studies. The concentration of α-tocopherol in breast milk was determined by its concentration in the serum. It is important to highlight that the intake of α-tocopherol was entered as an independent variable in the multiple regression analysis matrix. However, the model did not adjust, and significance was reached only after dietary data were excluded from the analysis matrix.

### 4.2. Vitamin A and E, and Carotenoids Distribution among Serum Lipoprotein Fractions Was Stable from Early- to Mid-Lactation

Highly nonpolar carotenes partition more easily into lipoprotein particles that are richer in neutral lipids (VLDL+LDL), and the more polar xanthophylls partition into HDL particles [37]. Indeed, our results showed that approximately 68% of serum lutein+zeaxanthin was in the HDL fraction in both early- and mid-lactation and serum carotenes were predominant in the serum fraction of LDL+VLDL. An important quality control of the protocol used for the differential precipitation of serum lipoproteins was the analysis of HDL-c in whole serum and in the HDL fraction, after precipitation of LDL+VLDL. There was no significant difference between HDL-c concentration in whole serum and in this lipoprotein fraction, indicating that there was no co-precipitation of HDL with LDL+VLDL. Retinol was evenly distributed between HDL and non-HDL serum fractions and was unchanged from early- to mid-lactation, consistently with previous reports [3]. However, the protocol used herein to precipitate LDL+VLDL does not allow the discrimination between free retinol, transported bound to retinol binding protein (RBP), and the HDL fraction. It is worth investigating in the future whether RBP-mediated retinol transport varies during lactation. γ-Tocopherol was evenly distributed in the serum fractions of HDL and LDL+VLDL. In contrast, α-tocopherol was more concentrated in the LDL+VLDL fraction, possibly because vitamin E is transferred to HDL, and then promptly transfers into all other circulating lipoproteins [5], which possibly favors its uptake by the mammary glands and transfer into breast milk.

In summary, the distribution of fat-soluble vitamins among plasma lipoprotein fractions, HDL and LDL+VLDL, in lactating women was similar to that of non-pregnant non-lactating women [37], despite the high adipose tissue turnover that is common to exclusive lactation. In addition, this distribution pattern was stable from early- to mid-lactation in the present study. From data in Figure 1, we estimated the sample size required to assess potential differences between early- and mid-lactation in the distribution of vitamin A and E, and carotenoids among HDL and LDL+VLDL fractions (Appendix A). Accordingly, it may be speculated that for the carotenes (α- and β-carotene and lycopene) and for retinol, it is unlikely that variations in their distribution between the lipoprotein fractions would be reasonably detectable—or that if detectable, they would have a biological role. Therefore, in the case of retinol and β-carotene, it is very unlikely that the decrease in the concentrations in breast milk, until mid-lactation, is related to their distribution between HDL and LDL+VLDL serum fractions. However, changes in the distribution of the xantophylls (Lut + Zea) and tocopherols (α and γ) in serum fractions might be detected between early- and mid-lactation with sample sizes of approximately 100 and 500 depending on the nutrient (Appendix A) and, in the case of α-tocopherol and the xantophylls, this metabolic distribution among serum lipoproteins might have a role in the decreasing transfer to breast milk from early- to mid-lactation, if confirmed in future studies. The major limitation of this work is the limited number of subjects, that limited reaching a significance level in some analyses and the capacity to extrapolate the results found to larger population groups. However, our results indicate new potentially interesting points to be investigated in the future, which might improve nutrition guidance for lactating women, especially if at high risk of sub-adequate intake of vitamin E.

## 5. Conclusions

Despite constant serum levels, the content of vitamin A and E in breast milk decreased throughout lactation, imposing risks of deficiency to the breastfed infants. Subclinical vitamin A deficiency in the lactating mothers was undetectable by serum retinol concentration in early- or mid-lactation. On the other hand, dietary assessment by the MSM was fairly sensitive to detect vitamin A inadequacy and might be used in the future to identify population groups at risk and guide maternal dietary advice on food choices or eventually supplementation in order to prevent sub-adequate nutritional status in breastfed newborns. A similar trend was observed for vitamin E, because breast the milk concentration of α-tocopherol seemed largely more sensitive than its serum concentration to the sub-adequate intake of the vitamin. Future prospective clinical trials providing nutritional advice to lactating mothers with sub-adequate vitamin A and E intake and the assessment of breast milk levels and infant nutritional status of these vitamins would contribute to widen the impact of this research.

## Figures and Tables

**Figure 1 nutrients-11-02025-f001:**
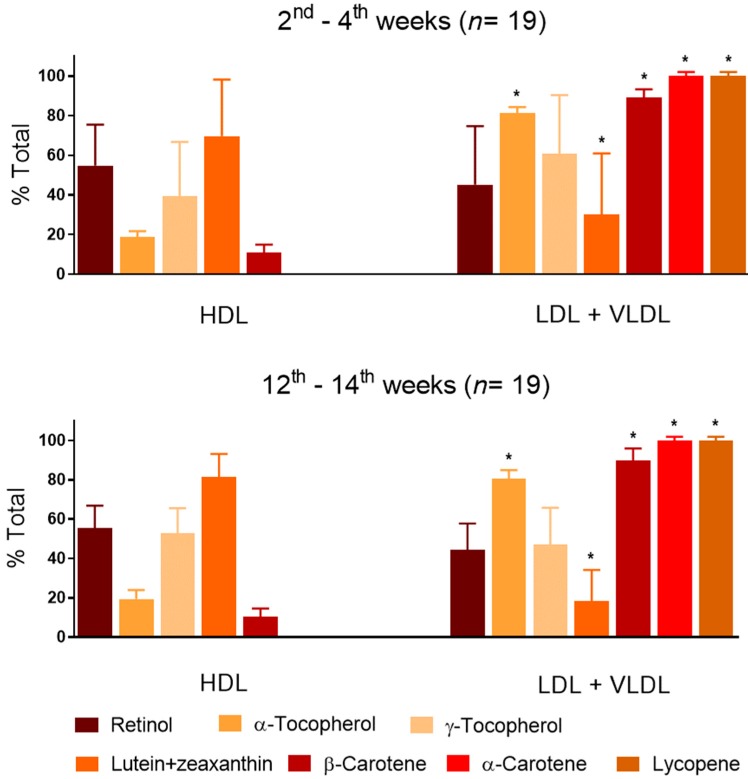
Distribution of fat-soluble vitamins in blood serum lipoprotein fractions in early- and mid-lactation. * Significantly different from the contents (%) in the HDL fraction (paired *t*-test; *p* < 0.05). Error bars represent standard deviations.

**Table 1 nutrients-11-02025-t001:** General characteristics of adult Brazilian lactating women in early- (2nd to 4th week) and mid-lactation (12th to 14th week) (*n* = 19).

Characteristics	Lactation Period		*p*	Recommended Value or Range ^1^
	Early Lactation 2nd–4th Week	Mid-Lactation 12th–14th Week		
Body mass index (kg/m^2^)	25.9 ± 4.0	25.3 ± 4.2	0.068 *	18–24.9 [6]
Hemoglobin (g/dL)	11.6 ± 1.7	12.5± 1.6	0.029 *	≥12.0 [26]
Hematocrit (%)	39.2 ± 3.5	37.8 ± 3.5	0.131	≥36 [26]
Triglycerides (mg/dL)	74.7 ± 21.4	57.0 ± 21.2	0.007 *	<150 [27]
Total cholesterol (mg/dL)	190.2 ± 41.9	182.2 ± 41.0	0.219	<200 [27]
LDL-c (mg/dL)	156.0 ± 44.1	146.6 ± 35.9	0.164	100–129 [27]
HDL-c (mg/dL)	34.2 ± 6.2	35.6 ± 9.4	0.601	>60 [27]

Data presented as the mean ± standard deviation; * Significant differences between lactation periods, paired *t*-test. ^1^ Shown in the same respective units as the first column.

**Table 2 nutrients-11-02025-t002:** Intake of energy and nutrients by Brazilian nursing women from the 2nd to the 14th week of lactation.

Energy and Nutrients	Intake (24hR-MSM)	Inadequacy (%) ^3^	Nutrient Intake Adequacy	
			Reference Value (Daily Intake) ^4^	Method [Ref.]
Energy (kcal)	1577 ± 164 ^1^	53	—	BMI distribution [14]
Carbohydrate (en%)	51.2 ± 7.7 ^1^	21	160 g ^5^	EAR cut-off [14]
Protein (en%)	17.7 ± 3.1 ^1^	47	1.05 g/kg ^5^	EAR cut-off [14]
Total lipids (en%)	31.1 ± 0.52 ^1^	26	20 to 35	AMDR [14]
Vitamin A (μg RE)	824.7 ± 21.8 ^1^	74	900	EAR cut-off [13]
β-Carotene (μg)	3249 (1408–6707) ^2^	—	3000–6000 [12]	—
α-Carotene (μg)	1053 (56–3712) ^2^	—	—	—
Lycopene (μg)	1854 (302–6472) ^2^	—	≥5000 [12]	—
Lutein + zeaxanthin (μg)	2446 (872–4873) ^2^	—	—	—
Vitamin E (mg)	4.4 ± 0.9 ^1^	100	16	EAR cut-off [12]

^1^ Data expressed as the mean ± standard deviation; ^2^ Median (minimum–maximum); ^3^ Estimated frequency of volunteers with inadequate intake; ^4^ based on estimated average requirement values whenever available, and are in the same units as the first column, except stated otherwise; in the case of carotenoids (β-carotene and lycopene), the reference value was taken as the daily intake level considered prudent for maintaining health; ^5^ EAR is used to assess adequacy of carbohydrate and protein intake, but usual intake expressed as en% are, respectively 45–65 and 5–10 [14]. AMDR: acceptable macronutrient distribution ranges; en%: macronutrient intake as a percentage of energy intake; 24hR-MSM: two 24-h recalls were used, and data were assessed by the Multiple Source Method; BMI: body mass index; EAR: Estimated Average Requirement.

**Table 3 nutrients-11-02025-t003:** Concentrations of fat-soluble vitamins (µmol/L) in the serum and breast milk of adult Brazilian nursing women (*n* = 19), in early- (2nd to 4th week) and mid- (12th to 14th week) lactation, and respective cut-off values for inadequacy ^1^.

Vitamin	Serum				Breast Milk			
	Lactation Period			Undernutrition Cut-Off Value	Lactation Period			Inadequacy Cut-Off Value
	2nd–4th Week	12th–14th Week	*p*		2nd–4th Week	12th–14th Week	*p*	
Retinol	1.50 ± 0.30	1.48 ± 0.31	0.913	0.7 [28]	2.3 ± 0.78	2.0 ± 0.72	0.013 *	1.05 [1,28]
β-Carotene	0.61 ± 0.07	0.59 ± 0.05	0.628	—	0.17 ± 0.02	0.14 ± 0.3	0.001 *	—
α-Carotene	0.43 ± 0.09	0.44 ± 0.08	0.720	—	0.04 ± 0.01	0.03 ± 0.01	0.304	—
Lycopene	0.20 ± 0.03	0.18 ± 0.04	0.530	—	0.04 ± 0.01	0.04 ± 0.00	1.000	—
Lutein + zeaxanthin	0.24 ± 0.04	0.23 ± 0.07	0.633	—	0.07 ± 0.02	0.06 ± 0.01	0.026 *	—
α-Tocopherol	11.1 ± 1.11	10.1 ± 1.23	0.089	11.6 [12]	1.29 ± 0.25	1.07 ± 0.13	0.001 *	7.4 [28]
γ-Tocopherol	0.76 ± 0.15	0.75 ± 0.13	0.896	—	0.38 ± 0.04	0.35 ± 0.04	0.436	—

^1^ Data expressed as the mean ± standard deviation. * Significant differences between lactation periods, paired *t*-test.

**Table 4 nutrients-11-02025-t004:** Multiple regression models of the concentrations of vitamin A and E (μmol/L) in the milk of Brazilian nursing women (*n* = 19), to assess its associations with the intake ^1^ and serum contents of these vitamins.

Model N	Dependent Variable *Breast Milk* ^2^	Independent Variables *Serum* ^A^ and *Diet* ^B^	Coefficients		Weight in the Model ^3^	Adj. *R* ^2^	Error (%) ^4^	*p* ^5^
			Value	*p*				
Early-Lactation: 2nd–4th Week								
1	Retinol	Vitamin A ^B^	5.33 × 10^−3^	<0.0001	100%	94.9	9%	<0.0001
2	β-Carotene	β-Carotene ^A^	2.25 × 10^−1^	0.0045	63%	87.7	14%	<0.0001
		β-Carotene ^B^	2.52 × 10^−5^	0.0430	37%			
3	α-Tocopherol	α-Tocopherol ^A^	1.24	<0.0001	100%	96.2	18%	<0.0001
Mid-Lactation: 12th–14th Week								
4	Retinol	Vitamin A ^B^	4.79 × 10^−3^	<0.0001	100%	94.7	14%	<0.0001
5	β-Carotene	β-Carotene ^A^	2.34 × 10^−1^	<0.0001	100%	78.4	17%	<0.0001
6	α-Tocopherol	α-Tocopherol ^A^	1.03	<0.0001	100%	97.9	12%	<0.0001

^1^ Data from the average of two 24-h recalls corrected by the Multiple Source Method (Vitamin A, μg RE/day; β-Carotene, μg/day); ^2^ Concentrations of fat-soluble vitamins in milk; ^3^ Weight in the model = independent variable coefficient × variable average content in samples; ^4^ Relative error of estimate = (estimated absolute error × 100%)/average value of the dependent variable; ^5^ Model significance. Superscript letters indicate whether the vitamins included as independent variables in the model were serum concentration (^A^) or dietary intake (^B^). Variables included in the starting analysis matrices before running the backward stepwise regression analysis (*p*-to-remove ≥ 0.05): model 1, vitamin A intake (RE), serum retinol and fat intake; model 2, β-carotene intake and serum β-carotene; model 3, vitamin E intake and serum α-tocopherol; model 4, vitamin A intake (RE), serum retinol and fat intake; model 5, β-carotene intake and serum β-carotene; model 6, vitamin E intake and serum α-tocopherol. Adj. *R*^2^: adjusted coefficient of determination for the fitted model.

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
