# Peer review of "Breast Milk Content of Vitamin A and E from Early- to Mid-Lactation Is Affected by Inadequate Dietary Intake in Brazilian Adult Women"

_nutrients, 2019, doi:10.3390/nu11092025_

Round 1

Reviewer 1 Report

Please read the document attached

Author Response

We thank the reviewer's appreciation and careful revision of our work. Please find in the attached document our point-by-point responses to the points raised during revision.

Reviewer 2 Report

Dear Author,

I consider the focus of the study really interesting.

However I have some observations on the methods and on the discussion of the draft;

Methods:

-given the many variables evaluated, it is necessary to express calculations for the power of the study and the required minimum sample size? 

-why was it chosen to analyze only mature milk?

-the first group of milk samples (2-4 weeks) include a time range very ample. Human milk expressed 2 weeks after birth is considered still transitional milk. Why was this choice of time range made?

-why wasn't the sample collected after completely emptying a breast?

- Were the samples protected from light to prevent a possible vitamin loss?

Discussion:

-I would increase the paragraphs of discussion of the results in comparison to the data reported in the literature (regarding the same and/or different population).

- in the end, I would speculate on the possible effect on the clinical practice of this study.

Author Response

(The authors gave the same response as above.)

Reviewer 3 Report

I have carefully read the manuscript entitled ,,Breast milk content of vitamin A and E from early- to 2 mid-lactation is affected by inadequate dietary intake 3 in Brazilian adult women,,

Importance of the research is good. Authors give data about vitamin A and E content in the breast milk seemed to be more sensitive than their respective blood concentrations, and dietary assessment by the MSM method.  The manuscript is interesting and well written. Methods, style & overall representation are correct. References are correctly presented. Work includes current publications on the subjects.

Author Response

We thank the reviewer's appreciation and careful revision of our work.

Round 2

Reviewer 1 Report

I thank authors to have taken into consideration my comments.

Please suppress the units in columns that you added in Tables 1 and 2 and homogenize them with those in column 1.

Author Response

"I thank authors to have taken into consideration my comments.

Please suppress the units in columns that you added in Tables 1 and 2 and homogenize them with those in column 1."

     We appreciate the reviewer's careful revision of our manuscript. Please see that Tables 1 and 2 in the revised manuscript were mended as suggested. The exception being the reference value for intake (EAR) of carbohydrates and proteins (Table 2) that are not expressed as AMDR (en%). EAR (g/day and g/kg/day) is the way that the IOM express the reference values for intake of these nutrients. We chose to show data as AMDR because this is the most common form to represent intake of macronutrients. In order to compromise the reviewer's suggestion with this common practice in the area of nutrition and dietetics, the Table 2 in the revised manuscript shows the recommended AMDR for these two macronutrients (in Table notes), making it more directly comparable to the data we show in column 2 in this Table.

Reviewer 2 Report

I don't have further suggestions

Author Response

"I don't have further suggestions"

     We thank the reviewer for his/her careful revision and appreciation of our work.